# UNIFYING STRUCTURAL PROXIMITY AND EQUIVALENCE FOR ENHANCED DYNAMIC NETWORK EMBEDDING

## ABSTRACT

Dynamic network embedding methods transform nodes in a dynamic network into low-dimensional vectors while preserving network characteristics, facilitating tasks such as node classification and community detection. Several embedding methods have been proposed to capture *structural proximity* among nodes in a network, where densely connected communities are preserved, while others have been proposed to preserve *structural equivalence* among nodes, capturing their structural roles regardless of their relative distance in the network. However, most existing methods that aim to preserve *both* network characteristics mainly focus on static networks and those designed for dynamic networks do not explicitly account for inter-snapshot structural properties. This paper proposes a novel unifying dynamic network embedding method that simultaneously preserves both structural proximity and equivalence while considering inter-snapshot structural relationships in a dynamic network. Specifically, to define structural equivalence in a dynamic network, we use temporal subgraphs, known as dynamic graphlets, to capture how a node's neighborhood structure evolves over time. We then introduce a temporal-structural random walk to flexibly sample time-respecting sequences of nodes, considering both their temporal proximity and similarity in evolving structures. The proposed method is evaluated using five real-world networks on node classification where it outperforms benchmark methods, showing its effectiveness and flexibility in capturing various aspects of a network.

## 1 INTRODUCTION

Network embedding transforms graph nodes into low-dimensional vectors while preserving network characteristics. These embeddings serve as inputs for tasks like link prediction, node classification, community detection, and graph visualization (Goyal & Ferrara, 2018; Cui et al., 2018). As real-world networks often change over time (Xue et al., 2022), dynamic network embedding is essential to capture this evolving nature. Developing effective node embedding methods requires consideration of various structural network properties.

One fundamental network characteristic is structural proximity. Studies have shown that nodes closer in a network often share similar properties or functions. For instance, in protein-protein interaction networks, nearby proteins typically share functions or are part of the same metabolic pathway (De Las Rivas & Fontanillo, 2010; Durek & Walther, 2008). In social networks, individuals tend to connect with others who are similar in demographics, backgrounds, or interests (Block & Grund, 2014).

Structural equivalence, on the other hand, focuses on the similarity between nodes based on their roles or functions within the network, regardless of their proximity. Two nodes are structurally equivalent if they share similar connection patterns. This concept helps identify consistent patterns and roles across different parts of the network. For example, users acting as mediators across a social network, who connect different communities together, might not belong to the same community but instead share similar connection patterns with other users who also act as mediators. Structural equivalence is evident in real-world networks, including social networks (Lorrain & White, 1971; Charbey & Prieur, 2019), transportation networks (Bai et al., 2021), and biological networks,

e.g. protein-protein interaction networks (Milenković & Pržulj, 2008; Davis et al., 2015) and brain networks (Finotelli et al., 2021).

## 2 RELATED WORK

### 2.1 STATIC NETWORK EMBEDDING

Various methods have been proposed to preserve structural proximity in static networks. Traditional methods include LLE (Roweis & Saul, 2000) and Laplacian eigenmaps (Belkin & Niyogi, 2001). Matrix factorization-based methods, such as HOPE (Ou et al., 2016), preserve higher-order proximities using a high-order proximity matrix and singular value decomposition, while GraRep (Cao et al., 2015) captures structural proximities across different neighborhood sizes using $k$-step probabilities. Random walk-based methods like DeepWalk (Perozzi et al., 2014) and node2vec (Grover & Leskovec, 2016) extract node neighborhoods with random walks and learn embeddings via a Skipgram (Mikolov et al., 2013a) model. Deep learning-based methods, such as SDNE (Wang et al., 2016), use a deep learning framework to maintain both local and global network structures.

In addition to methods focusing on structural proximity, several static network embedding methods address structural equivalence and capture node roles within the network. struc2vec (Ribeiro et al., 2017) captures structural similarities using node degrees, constructing a multi-layer graph where each layer encodes different resolutions of structural similarity. Random walks are applied, and Skip-gram is used to learn the final embeddings. GraphWave (Donnat et al., 2018) uses heat wavelets to describe neighborhood structures, effectively summarizing local patterns around each node. Another method (Wang et al., 2020) proposes structural role embedding in hyperbolic space. Additionally, Ahmed et al. (2020) performs attributed random walks to learn role-based embeddings.

### 2.2 DYNAMIC NETWORK EMBEDDING

Various dynamic network embedding methods build upon static methods to capture structural proximity in time-varying networks. CTDNE (Nguyen et al., 2018) extends DeepWalk by using temporal random walks to generate temporal node sequences, capturing temporal dynamics in node embeddings. Other methods include T-EDGE (Lin et al., 2020), which considers weighted networks, and tNodeEmbed (Singer et al., 2019), which builds on node2vec and employs LSTM (Hochreiter & Schmidhuber, 1997) for evolving interactions. De Winter et al. (2018) extends node2vec for dynamic link prediction. Pandhre et al. (2018) learns embeddings from random walks within the same snapshots as well as temporal walks across different snapshots to capture spatio-temporal dynamics.

Regarding structural equivalence in dynamic networks, Wang et al. (2021) first explores structural roles by extending the idea from struc2vec to dynamic networks. $k$-hop neighborhoods structural distance is calculated for node pairs at each timestep and aggregated to form a historical structural distance. However, this method does not consider structural dynamics between timesteps, as structural information is extracted independently at each timestep. Some works focus on a specific kind of structure, a triad, and model its evolution process. Zhou et al. (2018) proposes DynamicTriad which preserves the evolution pattern of a triad by modeling the triadic closure process to get node embeddings for each time snapshot. Huang et al. (2020) further proposes MTNE that models the triad evolution using Hawkes process (Hawkes, 1971).

### 2.3 UNIFYING STRUCTURAL PROXIMITY AND EQUIVALENCE

Several static network embedding methods aim to capture both structural proximity and equivalence simultaneously. Lyu et al. (2017) proposes a node embedding method that uses local subgraphs, or graphlets, to measure structural equivalence among nodes, in addition to node neighborhood information. However, this method limits structural similarity to nearby nodes in the $S$th-order neighborhood. Shi et al. (2019) considers graphlet-based structural equivalence between all node pairs and introduces joint representation learning to preserve both structural proximity and equivalence. Shi et al. (2021) defines structural equivalence using the graphlet degree vector (GDV) and employs cross-layer random walks on the original and structural similarity networks to capture both proximity and equivalence.

For dynamic networks, Liu et al. (2020) proposes a $k$-core based temporal Graph Convolution Network (CTGCN) that preserves both nodes' connective proximity and global structural similarity based on $k$-core (Nikolentzos et al., 2018) subgraphs or maximal subgraphs where all nodes have a degree of at least k. Node features are propagated along the k-core subgraphs within each time snapshot where RNN is then utilized to model the temporal dependency at different timestep. In a similar manner, Li et al. (2024) proposes a temporal Graph Convolution Network based on k-truss (Cohen, 2008) subgraphs (TTGCN) defined based on triangles. However, the topological structure considered to be preserved for both methods is defined within each time snapshot independently, as k-core and k-truss network structures are considered independently at each timestep.

# 3 OUR CONTRIBUTIONS

Majority of network embedding methods aim to capture either proximity or equivalence, but not both. Furthermore, existing methods that aim to preserve both characteristics are mostly designed for static networks, while the methods for dynamic networks do not explicitly consider inter-snapshot structural relationships. To address these limitations, we are motivated to propose a unifying network embedding framework that preserves both structural proximity and equivalence in a dynamic network, while considering inter-snapshot structural relationships. The main contributions of our work can be summarized as follows:

- We propose a unifying network embedding framework which preserves both structural proximity and equivalence in a dynamic network. We construct a structural similarity network based on dynamic graphlets (Hulovatyy et al., 2015), which explicitly accounts for inter-snapshot structural dynamics.

- We propose a temporal-structural random walk to generate temporal node contexts that capture both temporally close nodes and nodes with similar structural roles. We introduce the $\alpha$ hyperparameter that can be tuned to capture different degrees of task-specific network characteristics, offering flexibility and interpretability to the task of network learning.

- We evaluate the proposed method on five real-world networks for node classification and compare it with five benchmark methods. Our method outperforms the benchmarks. Additionally, we are able to infer the importance of each network characteristic across different networks using the $\alpha$ hyperparameter.

# 4 PROBLEM STATEMENT AND PRELIMINARIES

**Proximity and Equivalence Preserving Dynamic Network Embedding.** Given a dynamic network, $G = (V, E_T)$ where $V$ is the set of nodes shared across all timesteps and $E_T \subseteq V \times V \times T$ is the set of all dynamic edges and $E_t \in E_T$ is the edge at timestep $t$. The dynamic network embedding aims to learn a mapping function for time-respecting embedding $f : V \to \mathbb{R}^d$, where $d$ is the embedding dimensions and $d \ll |V|$. Additionally, the node embeddings $\mathbf{v}_i$ and $\mathbf{v}_j$ preserves both structural proximity and equivalence of node $v_i$ and $v_j$ in $G$.

## 4.1 STRUCTURAL PROXIMITY

Structural proximity in a dynamic network refers to how closely connected nodes are to each other over time. Preserving proximity is useful in preserving communities where nodes in close proximity are considered to be in the same community and share the same characteristics. Specifically, an edge $e_{ij}^t = (v_i, v_j) \in E_t$ between nodes $i$ and $j$ indicates first-order proximity between the two nodes at time $t$.

## 4.2 STRUCTURAL EQUIVALENCE

Structural equivalence identifies roles of nodes across different parts of the network based on their structural patterns. Regardless of their distance, a group of nodes can be considered structurally equivalent if they have the same local structures, e.g., mediator users across the network who connect different communities together. This concept is formally introduced in this section.

Graphlets and graphlet degree vector (GDV) have been widely used to capture local topological structure of nodes in real world networks (Sarajlić et al., 2016; Charbey & Prieur, 2019; Finotelli et al., 2021; Milenković & Pržulj, 2008). Graphlets are small, non-isomorphic, induced subnetworks. Nodes within the same graphlet are said to be of the same *automorphism orbit*, if they can be mapped to one another by automorphism, or in simple terms, have identical connection patterns. Figure 1 (a) shows all graphlets with up to four nodes where for each graphlet, its automorphism orbits are denoted in unique colors. The graphlet degree vector (GDV) of a node, serving as its topological signature, is a vector where each element corresponds to the number of times the node takes part in a specific orbit of a graphlet.

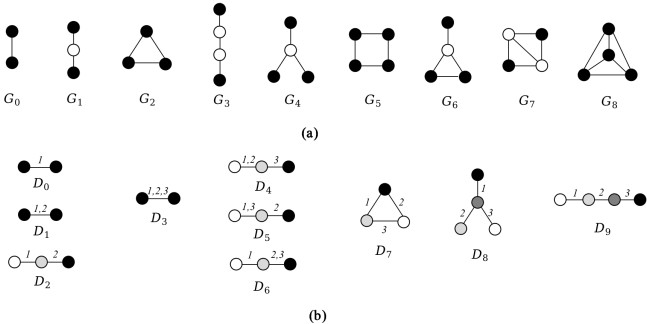

Figure 1: (a) All static graphlets with up to four nodes. Each graphlet has its unique automorphism orbits denoted in different colors, e.g. $G_0$ has one unique orbit, shown in black, while $G_3$ has two unique orbits shown in black and white. (b) All dynamic graphlets with up to three events. Automorphism orbits are shown in different colors. Ordered events of each graphlets are labeled with numbers where multiple events can occur on the same edge as labeled by numbers separated by commas.

To quantify the structural equivalence among nodes in dynamic networks, we use a dynamic generalization of graphlets, namely *dynamic* graphlets and *dynamic* GDV (D-GDV) (Hulovatyy et al., 2015). Dynamic graphlets introduce temporal information to the edges of graphlets, identifying each edge in a graphlet with a specific order in time. Hence, the D-GDV of a node, which summarizes the involvement of the node in different dynamic graphlet orbits, provides a topological dynamics signature of that node. Furthermore, since each edge in a dynamic graphlet can correspond to a different timestep, as we will later elaborate, it explicitly accounts for inter-snapshot dynamics in the network. The definitions of dynamic graphlet and dynamic GDV (D-GDV) (Hulovatyy et al., 2015) are as follows:

*Definition 3.3.1* $\Delta t$-**time respecting path**. Nodes $s$ and $d$ are said to be connected by a $\Delta t$-*time respecting path*, if there is a sequence $(v_0, u_0, t_0, \sigma_0), (v_1, u_1, t_1, \sigma_1), \ldots, (v_k, u_k, t_k, \sigma_k)$ such that $v_0 = s, u_k = d, \forall i \in [0, k-1] u_i = v_{i+1}$ and $t_{i+1} \in [t_i + \sigma_i, t_i + \sigma_i + \Delta t]$ or intuitively there is a temporal path from $s$ to $d$.

*Definition 3.3.2* A temporal network is called $\Delta t$-**connected** if for any pair of nodes, there is a $\Delta t$-time respecting path between the two nodes.

*Definition 3.3.3* **Dynamic graphlets**. Isomorphic $\Delta t$-connected temporal subgraphs, where two $\Delta t$-connected temporal subgraphs correspond to the same dynamic graphlet if they are topologically identical and their events occur in the same order. Figure 1 (b) shows all dynamic graphlets with up to three events, where the order of events is labeled along the edges and automorphism orbits for each graphlet are denoted in unique colors.

*Definition 3.3.4* **Dynamic graphlet degree vector (D-GDV)** of a node is a vector where each element corresponds to the number of times the node takes part in a specific orbit of a dynamic graphlet. This summarizes the dynamic graphlet involvement of a node and serves as its topological signature.

### 4.2.1 DYNAMIC GRAPHLET-BASED STRUCTURAL EQUIVALENCE

For our work, we define structural equivalence in a dynamic network based on dynamic graphlets. Given the dynamic graphlet degree vectors (D-GDVs) of all nodes in the network, the structural equivalence of two nodes $v_i$ and $v_j$, is based on the Euclidean distance of their D-GDVs in the PCA (Abdi & Williams, 2010) space. Specifically, the structural equivalence $s_{ij}$ is defined as

$$s_{ij} = \frac{1}{1 + d(v_i, v_j)},$$  (1)

where $d(v_i, v_j) = \|DGDV'(v_i) - DGDV'(v_j)\|$ is the distance between the two D-GDVs in the PCA space. A weighted dynamic graphlet-based structural similarity network, $S = (V, E_S)$, is then constructed where each weighted edge $e_{ij} = (v_i, v_j, s_{ij}) \in E_s$ represents the structural equivalence $s_{ij}$ between nodes $v_i$ and $v_j$. The network is made sparse by having each node keep only the top $k$ most similar neighbors. In this network $S$, structural proximity translates to structural equivalence in the original network $G$.

## 5 METHODOLOGY

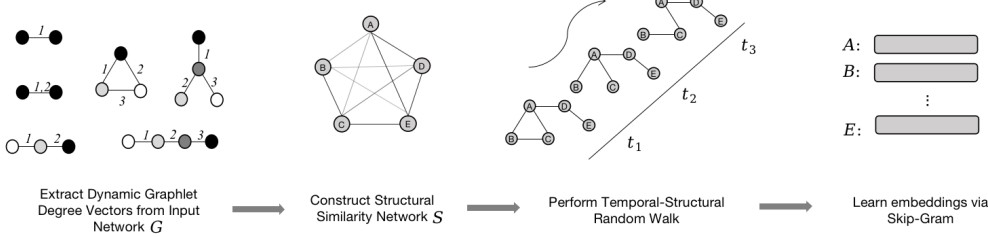

Extract Dynamic Graphlet Degree Vectors from Input Network $G$   →   Construct Structural Similarity Network $S$   →   Perform Temporal-Structural Random Walk   →   Learn embeddings via Skip-Gram

Figure 2: The proposed methodology workflow.

### 5.1 TEMPORAL-STRUCTURAL RANDOM WALK

To capture proximity and equivalence in a dynamic network, we introduce a temporal random walk algorithm, called temporal-structural random walk, that samples a time-respecting sequence of nodes that includes nodes that are temporally close in $G$ and nodes that are structurally similar based on dynamic graphlets, i.e., nodes that are close together in $S$.

We first describe a temporal random walk. A temporal random walk is a variation of the traditional random walk, adapted for temporal networks, which introduces the dimension of time into its transitions. The resulting walk reflects the temporal order of events in the network. Formally, a temporal walk in a dynamic network $G$ is a sequence of nodes $\{v_1, v_2, ..., v_l\}$ where the edge $e_i$, which is the edge connecting $v_i$ and $v_{i+1}$ for $1 \leq i < l - 1$, satisfies $\mathcal{T}(e_i) \leq \mathcal{T}(e_{i+1})$ where $\mathcal{T}(e_i)$ and $\mathcal{T}(e_{i+1}) \in 1 \dots T$ are the timesteps of edges $e_i$ and $e_{i+1}$, respectively. This incorporates time information into the walk and ensures that the walk is a forward move in time and that it is an ordered sequence of events in the network.

Given a random walker at node $v_i$ at time $t$, the temporal edge neighborhood for node $v_i$ at time $t$ is

$$\mathcal{N}_T(v_i, t) = \{e_{ik}^{t'} = (v_i, v_k, t') \in E_T \text{ where } t' \geq t\}.$$

The transition probability of walking an edge in the neighborhood is

$$\mathcal{P}_T(e_{ij}^{t'} | v_i, t) = \frac{\exp[t - t']}{\sum_{e \in \mathcal{N}_T(v_i)} \exp[t - \mathcal{T}(e)]}.$$  (2)

The exponential distribution is chosen to ensure the random walker is more likely to follow edges with smaller time gaps and avoid losing temporal information from bigger jumps in time.

Next, we introduce a structural random walk which samples structurally similar nodes that are not necessarily connected in the original network. The structural edge neighborhood $\mathcal{N}_S(v_i, t)$ for node $v_i$ is a set of weighted edges defined in $S$ connecting $v_i$ to its structurally similar nodes,

$$\mathcal{N}_S(v_i, t) = \{e_{ik}^t = (v_i, v_k, s_{ik}, t) \text{ where } (v_i, v_k, s_{ik}) \in E_S\}. \tag{2}$$

Unlike a temporal walk, since there is no notion of a temporal order of events when sampling two structurally similar nodes, the structural edge neighborhood for each node at time $t$ is defined only within the same time snapshot. However, the structural similarity weight $s_{ik}$, defined for each pair of nodes $v_i$ and $v_k$, considers the inter-snapshot dynamics through dynamic graphlets.

The transition probability of walking an edge in the structural neighborhood is then defined to be based on $S$ for all time snapshots $t$ as

$$\mathcal{P}_S(e_{ij}^t | v_i, t) = \frac{s_{ij}}{\sum_{e_{ik} \in \mathcal{N}_S(v_i)} s_{ik}}. \tag{3}$$

We introduce a hyperparameter $\alpha$ to control the importance weight between structural proximity (i.e., captured by the temporal walk) and equivalence (i.e., captured by the structural walk). Given a random walker at node $v_i$ at time $t$, the next edge $e$ will be sampled from the temporal neighborhood with probability $1 - \alpha$ and from the structural neighborhood with probability $\alpha$:

$$e \sim \begin{cases} P_T(.|v_i, t) & \text{with probability } 1 - \alpha, \\ P_S(.|v_i, t) & \text{with probability } \alpha. \end{cases} \tag{4}$$

When $\alpha = 0$, the walk captures only temporal proximity and reduces to a temporal random walk on the dynamic network $G$, which in this special case our method is equivalent to CTDNE (Nguyen et al., 2018). Conversely, when $\alpha = 1$, the walk considers only dynamic structural roles, equivalent to walking solely on the structural similarity network $S$.

## 5.2 Learning Node Embeddings via Skip-Gram Model

The Skip-gram (Mikolov et al., 2013a) model is used to learn node embeddings from the random walk sequences. The Skip-gram model is a three-layer neural network, originally used in natural language processing (NLP), which later has been widely adopted to learn node embeddings in networks.

Nodes in the network are treated as words, and the random walk sequences on the network are analogous to sentences in a text corpus. For each node $v_i$, the model maximizes the likelihood of finding the temporal context nodes appearing in the random walk sequences. More specifically, consider a node $v_i$ appearing in a random walk sequence, the sliding temporal context window of node $v_i$ is $\mathcal{W}(v_i) = \{v_{i-\omega}, ..., v_i, ..., v_{i+\omega}\}$ where $\omega$ is the window size and $\mathcal{T}(v_{i-\omega}, v_{i-\omega+1}) \le \cdots \le \mathcal{T}(v_{i+\omega-1}, v_{i+\omega})$. The likelihood of finding the context nodes conditioned on its embedding is as follows:

$$\log p(\mathcal{W}(v_i)|\mathbf{v_i}) = \sum_{v_j \in \mathcal{W}(v_i)} \log p(v_j \mid \mathbf{v_i}), \tag{5}$$

where $\mathcal{W}(v)$ represents the nodes around node $v$ in random walks. The probability $p(v_j \mid \mathbf{v_i})$, modeled using the softmax function, is

$$p(v_j \mid \mathbf{v_i}) = \frac{\exp(\mathbf{v}_j^T \mathbf{v}_i)}{\sum_{\mathbf{v}_k \in V} \exp(\mathbf{v}_k^T \mathbf{v_j})}. \tag{6}$$

As the sum of all nodes in the denominator $\sum_{\mathbf{v}_k \in V} \exp(\mathbf{v}_k^T \mathbf{v_j})$ can be computationally costly for large networks, negative sampling is generally adopted to reduce training running time (Mikolov et al., 2013b). Instead of summing over all the nodes, a small number of negative nodes (i.e., nodes that are not in the context window) will be sampled from a noise distribution. In this paper, we use a well-adopted modified unigram distribution. A negative node will be sampled from the distribution $P_n(x) = \frac{U(x)^{\frac{3}{4}}}{Z}$, where $U(x)$ is the unigram distribution of the frequency of node $x$ in all random walk sequences.

---

**Algorithm 1** Dynamic Network Embeddings

---

**Require:** an unweighted and (un)directed dynamic network $G = (V, E_T, \mathcal{T})$, number of walks $\beta$, context window size $\omega$, embedding dimensions $D$, maximum walk length $l$, maximum number of graphlet nodes $n$, maximum number of graphlet events $m$, number of neighbors to keep $k$
1: $S = \text{STRUCTURALSIMILARITYNETWORK}(G, n, m, k)$
2: Initialize set of walks $W = \{\}$
3: **for** iter = 0 to $\beta$ **do**
4:     $W_i = \text{TEMPORALSTRUCTURALWALK}(G, S, l)$
5:     Add $W_i$ to $W$
6: **end for**
7: $Z = \text{SKIPGRAM}(W, \omega)$

---

**Algorithm 2** Temporal-Structural Random Walk

---

**Require:** an unweighted and (un)directed dynamic network $G = (V, E_T)$, a weighted structural similarity network $S = (V, E_S)$, walk length $l$, balancing weight $\alpha$
1: Sample a starting edge in $G$, $(u, v, t)$
2: Initialize walk sequence $W = [u, v]$
3: Set $v_i = v$
4: **for** iter = 1 to $l - 1$ **do**
5:     $\mathcal{N}_T(v_i, t) = \{e_{ik}^{t'} = (v_i, v_k, t') \in E_T \text{ where } t' \geq t\}$
6:     $\mathcal{N}_S(v_i, t) = \{e_{ik}^t = (v_i, v_k, s_{ik}, t) \text{ where } e_{ik} = (v_i, v_k, s_{ik}) \in E_S\}$
7:     Define $\mathcal{P}_T(.|v_i, t)$ based on Equation (2)
8:     Define $\mathcal{P}_S(.|v_i, t)$ based on Equation (3)
9:     Sample new edge $e$ from $\mathcal{P}_T(.|v_i, t)$ with prob. $1 - \alpha$ and from $\mathcal{P}_S(.|v_i, t)$ with prob. $\alpha$ as in Equation (4)
10:    Set $v_i = Dst(e)$
11:    Set $t = \mathcal{T}(e)$
12:    Add $v_i$ to $W$
13: **end for**
14: **return** ramdom walk sequence $W$

---

**Algorithm 3** Structural Similarity Network $S$

---

**Require:** $G = (V, E_T)$, maximum number of graphlet nodes $n$, maximum number of graphlet events $m$, number of neighbors to keep $k$.
1: Compute Dynamic Graphlet degree vector $DGDV(v_i)$ for each node $v_i$
2: PCA decomposition $DGDV'$
3: **for** each node pair $i$ and $j$ **do**
4:     Compute structural similarity $s_{ij}$ according to (1)
5: **end for**
6: Initialize $E_S = \{\}$
7: **for** each node $v_i$ **do**
8:     Define an edge $(v_i, v_k, s_{i,k})$ for top $k$ largest weighted edges and add to $E_S$
9: **end for**
10: **return** $S = (V, E_S)$

---

## 6 COMPUTATIONAL COMPLEXITY ANALYSIS

### 6.1 TIME COMPLEXITY

Given the number of nodes $N = |V|$, the number of dynamic edges $M = |E|$, the embedding dimension $D$, the number of random walk sequences sampled per node $R$, the maximum length of a random walk $L$, and the maximum degree of a node $\Delta$, the time complexity of performing temporal random walks and learning Skip-gram model has been shown by Nguyen et al.

(2018) to be $\mathcal{O}\left(M + N\left(R\log M + RL\Delta + D\right)\right)$. For the number of dynamic graphlet types $S(n, m)$ which is a function of the number of $n$ nodes and $m$ events considered, $S(n, m) = \sum_{i=0}^{n-2} \frac{(-1)^{n+i}\binom{n-2}{i}(2i+1)^{m-1}}{2(n-2)!}$, $n \geq 3$, the time complexity of dynamic graphlets computation has been shown by Hulovatyy et al. (2015) to be $\mathcal{O}\left(M + M\left(\frac{S}{M}\right)^{m-1}\right)$. The time complexity of performing PCA on the dynamic graphlet degree vector with dimension $P$ using SVD is $\mathcal{O}\left(P^2 N\right)$, where $P \ll N$. The time complexity of constructing a structural similarity network is $\mathcal{O}\left(jN^2\right)$, where $j \ll N$ is the dimension kept after PCA. Therefore, the total running time is $\mathcal{O}(M + N(R\log M + RL\Delta + D) + M\left(\frac{S}{M}\right)^{m-1} + P^2 N + jN^2)$, which is affordable as generally it is sufficient to only consider a small number of nodes and events for dynamic graphlets. The specific number of nodes and events considered for dynamic graphlets used in our setting is specified in the *Experimental Settings* section.

## 6.2 SPACE COMPLEXITY

The space complexity for the temporal random walk is $\mathcal{O}\left(M + ND\right)$ (Nguyen et al., 2018). The space complexity for the structural random walk is $\mathcal{O}\left(kN\right)$, since each node requires storing the edge weights of top $k$ most structurally similar nodes. Therefore, the total space complexity is $\mathcal{O}\left(M + ND + kN\right)$.

## 7 EXPERIMENT AND RESULTS

In this section, we introduce the experimental settings and results. Our code with all datasets used in the experiment is publicly available at https://anonymous.4open.science/r/temporal-structural-walk-C7CC.

## 7.1 DATASETS

We use five real-world networks in our experiments: Hospital (Vanhems et al., 2013), Workplace (Génois et al., 2015), Enron (Carley, 1995), PPI-aging (Faisal & Milenković, 2014), and Brain [1]. Details of the datasets can be found in Appendix A.1.

## 7.2 EVALUATION METRICS AND BASELINE METHODS

We compare our method with five network embedding methods including DeepWalk (Perozzi et al., 2014), node2vec (Grover & Leskovec, 2016), struc2vec (Ribeiro et al., 2017), CTDNE (Nguyen et al., 2018), and D-GDV or dynamic graphlet degree vector (Hulovatyy et al., 2015) where PCA decomposition is applied for dimension reduction, with the dimensions kept so that 90% variance in the data remains.

The node embeddings from each method are used as inputs to a one-vs-rest logistic regression classifier for a node classification task where 5-fold cross validation is applied. Specifically, the model is trained on four parts and tested on the remaining part, this process is repeated five times with each part used exactly once as the test set, and the results are averaged to provide a performance estimate. The macro-average scores for Average Precision (AP) and Area Under the Receiver Operating Characteristic Curve (AUROC) are reported as mean and standard deviation obtained from the cross validation.

## 7.3 EXPERIMENTAL SETTINGS

The embedding dimension $d$ is set to 32 for all datasets. The maximum length of random walk $l$ is chosen from the set $\{10, 15, 20, 25, 30\}$ according to the length of the timesteps and sparsity of each network and is set to 25, 15, 20, 30, and 10, for Hospital, Workplace, Enron, PPI-aging, and Brain datasets, respectively. The context window size $\omega$ is set to 10 for all datasets. The hyperparameter

---

[1]https://tinyurl.com/y4hhw8ro

$\alpha$ is chosen over the interval $[0, 1]$ with 0.025 incremental search to get the optimal performance in terms of AP.

We use the constrained dynamic graphlets counting implementation provided by Hulovatyy et al. (2015) to compute D-GDV. According to their graphlet size and node classification performance analysis, increasing the number of nodes improves accuracy at a cost of higher computational complexity and a small graphlet size is shown to be effective. However, for a fixed number of nodes, increasing the number of events considered does not necessarily improve the performance. For the smallest network, Hospital, we consider dynamic graphlets with up to 6 events and 4 nodes. For the medium size network, Workplace and Enron, we consider dynamic graphlets with up to 4 events and 5 nodes. Lastly, for large networks, PPI-aging and Brain, we reduce the node size and consider dynamic graphlets with up to 4 events and 4 nodes. The structural similarity network $S$ is then constructed with the top $k$ similar neighbors kept for each node, where $k$ is set to be 5, 5, 5, 100, and 20, for Hospital, Workplace, Enron, PPI-aging, and Brain datasets, respectively.

## 7.4 NODE CLASSIFICATION IN DYNAMIC NETWORK

| Data Set | Algorithm | AP | AUROC |
|---|---|---|---|
| **Hospital** | D-GDV | $0.7535 \pm 0.0691$ | $0.8370 \pm 0.0943$ |
| | DeepWalk | $0.7054 \pm 0.0449$ | $0.8334 \pm 0.0275$ |
| | node2vec | $0.7782 \pm 0.0779$ | $0.8627 \pm 0.0727$ |
| | struc2vec | $0.5956 \pm 0.0374$ | $0.7326 \pm 0.0791$ |
| | CTDNE | $0.8407 \pm 0.0394$ | $0.9329 \pm 0.0403$ |
| | Ours | $\mathbf{0.8639 \pm 0.0710}$ | $\mathbf{0.9399 \pm 0.0527}$ |
| **Workplace** | D-GDV | $0.4713 \pm 0.1020$ | $0.6780 \pm 0.0782$ |
| | DeepWalk | $0.9755 \pm 0.0184$ | $0.9877 \pm 0.0133$ |
| | node2vec | $0.9766 \pm 0.0149$ | $0.9890 \pm 0.0103$ |
| | struc2vec | $0.4023 \pm 0.0405$ | $0.6363 \pm 0.0308$ |
| | CTDNE | $0.9836 \pm 0.0157$ | $0.9911 \pm 0.0090$ |
| | Ours | $\mathbf{0.9922 \pm 0.0077}$ | $\mathbf{0.9959 \pm 0.0043}$ |
| **Enron** | D-GDV | $0.3102 \pm 0.0216$ | $0.6624 \pm 0.0171$ |
| | DeepWalk | $0.4076 \pm 0.0433$ | $0.7780 \pm 0.0279$ |
| | node2vec | $0.4118 \pm 0.1152$ | $0.7499 \pm 0.0428$ |
| | struc2vec | $0.3118 \pm 0.0597$ | $0.6726 \pm 0.0460$ |
| | CTDNE | $0.4730 \pm 0.0807$ | $0.7967 \pm 0.0300$ |
| | Ours | $\mathbf{0.5145 \pm 0.0922}$ | $\mathbf{0.8047 \pm 0.0366}$ |
| **PPI-aging** | D-GDV | $0.2373 \pm 0.0363$ | $0.7576 \pm 0.0241$ |
| | DeepWalk | $0.2332 \pm 0.0615$ | $\mathbf{0.8418 \pm 0.0232}$ |
| | node2vec | $0.2375 \pm 0.0681$ | $0.8296 \pm 0.0191$ |
| | struc2vec | $0.2239 \pm 0.0561$ | $0.8226 \pm 0.0155$ |
| | CTDNE | $0.1035 \pm 0.0296$ | $0.7592 \pm 0.0322$ |
| | Ours | $\mathbf{0.2566 \pm 0.0311}$ | $0.7857 \pm 0.0262$ |
| **Brain** | D-GDV | $0.4544 \pm 0.0098$ | $0.8706 \pm 0.0043$ |
| | DeepWalk | $0.5111 \pm 0.0203$ | $0.9088 \pm 0.0033$ |
| | node2vec | $0.4956 \pm 0.0221$ | $0.9063 \pm 0.0048$ |
| | struc2vec | $0.1735 \pm 0.0034$ | $0.6422 \pm 0.0065$ |
| | CTDNE | $0.5571 \pm 0.0116$ | $0.9200 \pm 0.0019$ |
| | Ours | $\mathbf{0.5664 \pm 0.0156}$ | $\mathbf{0.9205 \pm 0.0027}$ |

Table 1: Comparison of algorithms on node classification tasks.

The proposed method is evaluated using node classification on five real-world networks. The results shown in Table 1 demonstrates the state-of-the-art performances of the proposed method. The networks exhibit varying degrees of proximity and equivalence characteristics that can be observed by the optimal tuned value of $\alpha$ for each network in Figure 3. The optimal $\alpha$ values for the majority of networks including Hospital, Workplace, Enron, and Brain, are found to be low in the range of 0.025 to 0.1, which indicates that the labels of the networks are more related to structural proximity than structural roles. However, the non-zero $\alpha$ values demonstrate that structural equivalence properties are beneficial for node classification. Specifically when compared to CTDNE (Nguyen et al., 2018), which is the special case of our model when $\alpha = 0$ (capturing only structural proximity), our model shows improvements in both AP and AUROC. The improvement is particularly large on the Enron and Hospital dataset, where our method has a relative improvement of 8.77% and 2.75% increase

in AP. The results also indicate significant improvements over static baselines such as DeepWalk, node2vec, and struc2vec, highlighting the temporal information captured by the proposed method.

On the other hand, the high optimal $\alpha$ value of 0.95 for the PPI-aging network indicates that node labels are more related to structural equivalence or structural roles. However, the optimal $\alpha$ value being less than one shows that proximity characteristics also contribute to the classification task. Given the highly imbalanced labels in the PPI-aging network, with a significantly larger negative class, AP is a more suitable metric for evaluation. Compared to struc2vec, which captures only structural roles, our proposed method achieves a relative improvement of 14.60% in AP. Furthermore, compared with D-GDV, whose structural information is used to construct our structural similarity network, our proposed method achieves a relative improvement of 8.13% AP, showing the enhanced performance from incorporating information on node proximity.

### 7.5 Sensitivity of the hyperparameter $\alpha$

The $\alpha$ hyperparameter, which controls the balance between node proximity and structural roles, introduces flexibility to the model as different tasks in the same network might require varying degrees of each network characteristic. Figure 3 shows the model performance in terms of AP across different datasets using different values of $\alpha$. The optimal value for $\alpha$ for each dataset tends to be either close to 0 or 1, indicating that each task primarily relies on either proximity or equivalence. However, there is an improvement when incorporating the minor characteristic to a small degree.

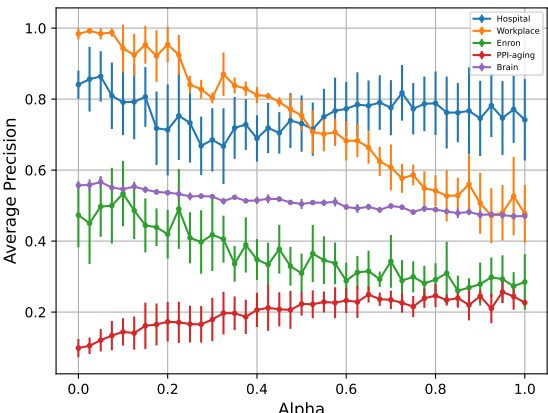

Figure 3: Average Precision for different values of hyperparameter $\alpha$, where the vertical lines represent the standard deviation.

## 8 Conclusion

In this work, we have studied the dynamic network embedding problem with the goal of capturing both structural proximity and equivalence of a dynamic network, while accounting for the inter-snapshot structural dynamics among nodes. We have quantified structural equivalence between two nodes in a dynamic network based on dynamic graphlets. A temporal-structural random walk method has been proposed to sample node sequences consisting of temporally close nodes and structurally similar nodes by introducing a hyperparameter $\alpha$ to balance the weight between the two network characteristics. The proposed method has demonstrated the state-of-the-art performances on five real-world datasets on node classification, capturing varying degrees of structural proximity and equivalence in dynamic networks. In this paper, the hyperparameter $\alpha$ was chosen by brute force search; for future work, a supervised strategy could be developed to learn the optimal $\alpha$ using node labels in the training set.

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

# A APPENDIX

## A.1 DATASETS

- Hospital Vanhems et al. (2013): A contact network between patients and health-care workers where the node labels are different individual roles (e.g., paramedical staffs, administrative staff, etc.).
- Workplace Génois et al. (2015): A contact network in an office building where the node labels are different workplace departments.
- Enron Carley (1995): An email communication network where the node labels are company roles (e.g., CEO, president, director, employee, etc.).

- PPI-aging Faisal & Milenković (2014): A dynamic age-specific protein-protein interaction network spanning 37 different ages, from 20 to 99 years. Node labels are binary, indicating whether the protein/gene is aging-related. The node labels are highly imbalanced with approximately 2% of the nodes in the positive class.
- Brain [2]: This dataset is obtained from functional magnetic resonance imaging (fMRI) data, where nodes represents cubes of brain tissue and edges between two nodes represent the similar degrees of activation at each time period. Node labels are brain functions (e.g., auditory processing, language processing, emotion processing, body movement, etc.).

Table 2: Dataset Detail

| Dataset | Nodes | Edges | Time Steps | Classes |
|---------|-------|-------|------------|---------|
| Hospital | 72 | 2,845 | 27 | 4 |
| Workplace | 92 | 9,827 | 20 | 5 |
| Enron | 182 | 9,880 | 45 | 7 |
| PPI-aging | 6,371 | 557,303 | 37 | 2 |
| Brain | 5,000 | 947,744 | 12 | 10 |

---

[2]https://tinyurl.com/y4hhw8ro

