# OpenReview forum: "Unifying Structural Proximity and Equivalence for Enhanced Dynamic Network Embedding"
_ICLR.cc/2025/Conference — ICLR 2025 Conference Withdrawn Submission_

### Official Review · Reviewer_QsTR · 2024-11-01

**Soundness:** 3
**Presentation:** 3
**Contribution:** 2
**Rating:** 3
**Confidence:** 2

**Summary:**

The paper proposes a novel random-walk-based node embedding approach for dynamic networks. The proposed framework relies on the generation of temporal graphlets which are used to capture the structural equivalence of nodes. The introduced random walking strategy uses the combination of temporal and designed structural node similarities while visiting nodes. The performance of the architecture is evaluated with respect to five chosen baseline methods for the node classification task.

**Strengths:**

- The paper is well-structured and clearly written.
- The authors introduce a novel random walk-based node embedding method and it outperforms the selected baselines.

**Weaknesses:**

- The paper has limitations in originality.
- The authors compare their model against non-recent methods, which weakens the impact of the reported improvements.
- Additionally, the computational complexity of the proposed method is quite high, making it potentially infeasible for large-scale networks.
- The experimental setup is not clear, specifically regarding hyperparameter selection for both the proposed and baseline models.

**Questions:**

The authors adapt the approach given in Subsection 4.2.1 to derive the structural equivalence of nodes, and they build a static network. However, in temporal networks, structural roles and equivalences of nodes might also change over time. It would be helpful for readers if the authors provided the motivation for why they preferred this method.

The reliance on temporal graphlets, as introduced in previous work (Hulovatyy et al., 2015), alongside a random-walking strategy that integrates DeepWalk and its extension to temporal networks, CTDNE, limits the originality of the proposed method.

It is unclear how the model parameters were tuned since the use of a validation set is not specified. Were parameters chosen based on training or test set performance?

The chosen baselines are not recent approaches. Authors should consider the more recent works such as [1,2]. Additionally, it is unclear whether the static approaches were run on the network as a whole by aggregating the link or on specific time slices, and whether baseline hyperparameters were optimized or left as defaults. Greater clarity on these aspects would strengthen the experimental section of the work.

Since node labels in dynamic networks may vary over time, the authors should clarify if they assume fixed labels across the timeline. The networks are also discrete-time and if the node labels are fixed, it would be interesting to explore the performance of the static approaches when they are run only by using the edges in the last time step or in the recent time steps.

The high time complexity of the proposed method raises concerns regarding its applicability to large-scale networks. The authors should discuss any strategies or future work aimed at addressing this limitation.

In Definition 3.3.1, the notation $t_i$ indicates the initial time of the edge but the meaning of $\sigma_i$ remains ambiguous. Clarifying this would help readers follow the notations in the paper.

In Lines 307-308, the authors state that $T(v_{i−\omega} ,v_{i−\omega+1}) ≤ ···≤T(v_{i+\omega}−1,v_i+\omega )$ but it might not always hold since we also sample from $P_S$.

- *[1] Rossi, Emanuele, et al. "Temporal graph networks for deep learning on dynamic graphs." arXiv preprint arXiv:2006.10637 (2020).*
- *[2] Çelikkanat, Abdulkadir, Nikolaos Nakis, and Morten Mørup. "Piecewise-velocity model for learning continuous-time dynamic node* representations." arXiv preprint arXiv:2212.12345 (2022).*

**Decision**

I recommend rejecting this paper due to my concerns about originality and experiments. While the paper presents a well-organized study, the proposed method heavily relies on existing techniques (temporal graphlets and adaptations from DeepWalk and CTDNE), which limits its contribution. Furthermore, the chosen baselines are not recent approaches, and some points regarding the parameter tuning process remain unclear, raising concerns about the validity of the reported performance improvements.

---

### Official Review · Reviewer_KHAp · 2024-11-01

**Soundness:** 3
**Presentation:** 3
**Contribution:** 2
**Rating:** 6
**Confidence:** 3

**Summary:**

This paper introduces a novel dynamic network embedding that unifies structural proximity and equivalence through dynamic graphlets and a temporal-structural random walk method. Structural proximity helps capture closely connected communities, while structural equivalence identifies nodes with similar structural roles regardless of their network position. The model's temporal-structural random walk algorithm enables sampling sequences that account for both temporal proximity and evolving structures, weighted by a tunable hyperparameter 𝛼, allowing task-specific flexibility. This embedding framework is evaluated on five real-world datasets for node classification, demonstrating improvements over existing benchmarks. The paper also provides a detailed analysis of time complexity, supporting the method’s computational efficiency, and shows sensitivity to 𝛼, highlighting the balance between structural proximity and equivalence.

**Strengths:**

- The paper provides clear motivation and well-structured explanations of methods, making it easy to follow.
- The unified framework effectively captures structural proximity and equivalence simultaneously, a novel feature in dynamic network embeddings.
- Time complexity analysis supports the model's computational feasibility
- The experimental results are well-organized, offering consistent improvements over benchmarks in node classification across diverse datasets.

**Weaknesses:**

- The current scope focuses primarily on node classification, limiting the model’s evaluation to a single task. A broader range of tasks, such as edge prediction, label prediction, or clustering, would better demonstrate the method's versatility.
- The approach works by alternating between temporal orders and structural equivalence. Still, it does not address their potential interactions or dependencies, which could be crucial for capturing complex network dynamics.
- The parameter 𝛼 is determined per dataset rather than per instance or graphlet, which may limit the model’s ability to capture instance-specific or graphlet-specific characteristics, potentially overlooking finer details.
- Although the method achieves performance gains, they are somewhat incremental compared to existing approaches, particularly on certain datasets.
- Despite referencing many relevant works, the authors compare their method with relatively outdated models only, missing recent state-of-the-art (SOTA) approaches. This limits the strength of the claims about effectiveness and generalizability.
- Comparisons on larger datasets, if feasible, could further validate scalability and performance robustness.

**Questions:**

- Could you elaborate on how the method would handle cases where temporal order and structural similarity interact or influence each other, and what impact this might have on embeddings?
- Have you considered varying α at a more granular level, such as per group, instance, or graphlet, to capture specific characteristics and enhance adaptability? If so, what are the challenges involved?
- Do you have plans to apply this model to tasks beyond node classification, such as edge prediction, label prediction, or clustering? These would provide additional insights into the flexibility of the approach.
- What are the barriers to evaluating this model on larger or more recent datasets, and would such testing reveal any scalability or performance limitations?

---

### Official Review · Reviewer_cs4Z · 2024-11-03

**Soundness:** 2
**Presentation:** 3
**Contribution:** 2
**Rating:** 3
**Confidence:** 4

**Summary:**

This paper introduces a dynamic network embedding method that encodes nodes as low-dimensional vectors while preserving both structural proximity (dense connections) and structural equivalence (similar structural roles) across time. Unlike existing methods focused on static networks or single-snapshot dynamics, this approach leverages "dynamic graphlets" to track neighborhood changes over time, using a temporal-structural random walk to sample nodes based on temporal proximity and structural similarity. Tested on five real-world networks, the method seems to perform better in node classification against competing methods.

**Strengths:**

- 1) The paper addresses a very interesting topic in graph representation learning, to learn low-dimensional embeddings that can capture both the structural proximity, as well as the structural equivalence. Importantly, they extend such effort the arduous case of dynamic graphs.

- 2) The paper is well written, and all ideas are communicated clearly enhancing the clarity of the manuscript.

- 3) The use of dynamic graphlets to construct a temporal-structural random walk is novel and interesting, while the introduction of the parameter alpha to control node proximity versus structural role learning adds valuable flexibility to the method

**Weaknesses:**

- 1) Missing discussion based on some key works in the topic of node proximity (homophily) versus structural equivalence. Especially in the work of Hoff (2007) the capturing of homophily and structural equivalence is achieved via the proximity metric chosen for expressing the node similarity. This essentially unifies the efforts adressed in this paper. It would be beneficial to the paper to also address/discuss such settings and how they are relevant to their method.

- 2) The proposed focuses on the temporal graph learning but the experimental set-up is focused on the static downstream task of node classification. There exist additional tasks the authors could consider like in Çelikkanat et al. (2022).

- 3) The. obtained results in the task of node classification provide in many cases marginal improvements that are within the standard deviations margins when compared to the other baselines. Additional results for significance should be performed like a Paired t-test or something similar.

- 4) The performance is based on the optimal alpha value, that if I am not mistaken is chosen based on the test set performance, introducing bias in the results.

- 5) Lack of artificial networks study; the optimal alpha value is found to be very small, as fast by the authors in their manuscript. Thus the importance of the method falls somehow short. The method should be established also in artificial networks that the setting can be controlled to show regimes where the alpha value gets a bigger role to successfully learn the node embeddings and roles.




Peter D. Hoff, Modeling homophily and stochastic equivalence in symmetric relational data (2007)

Çelikkanat et al., Piecewise-Velocity Model for Learning Continuous-time Dynamic Node Representations (2022)

**Questions:**

- What the effect of dimensionality in your proposed method? How sensitive is the performance in the choice of latent dimensions?

- The time complexity depends on $N^2$ is this the total size of the network of the size of the graphlet? If it is the size of total network the statement that the model complexity is affordable should be revised, as it scales quadratically with the number of network nodes.

---

### Official Review · Reviewer_CseL · 2024-11-05

**Soundness:** 2
**Presentation:** 4
**Contribution:** 1
**Rating:** 3
**Confidence:** 5

**Summary:**

The authors consider the problem of node embedding in dynamic networks. Two main types of embeddings are present in the literature: ones based on structural proximity (e.g., neighborhood-based embeddings) and ones based on structural equivalence (e.g., role-based embeddings). The authors propose a straightforward approach to capture both types into a single embedding using random walks on the temporal neighborhood and the structural neighborhood. They demonstrate improved node classification accuracy on 5 real network data sets.

**Strengths:**

- Proposed approach incorporates both elements of structural proximity and equivalence into a single embedding for dynamic networks, which has not been done before, to the best of my ability.
- Proposed approach seems to be technically sound.
- Paper is well written and easy to understand.

**Weaknesses:**

- Low novelty and significance. This is an incremental improvement to an area of research that is already well understood. If the paper gets rejected, I would suggest publishing in a venue that places less emphasis on significance, such as Transactions on Machine Learning Research.
- Weak baselines for comparison in the experiments. None of the baselines include both a structural proximity and equivalence component. A more useful comparison could be to a convex combination of a structural proximity embedding (e.g., CTDNE) + a structural equivalence embedding (e.g., D-GDV).
- No visualization or interpretation of embeddings is provided. This is also a weak point of the proposed single embedding combining structural proximity and equivalence--there is no ability to identify similar nodes according to each individual criterion.

**Questions:**

1. Why should a practitioner choose your proposed approach rather than just training both a structural proximity embedding and a structural equivalence embedding and then taking a convex combination of them? The weight in the convex combination could be tuned in the same way as the $\alpha$ parameter in your approach.
2. From your experiments, it appears that the structural equivalence-based embeddings perform much worse in node classification. Do you have any interpretation on why this is the case?

---

### Note · Authors · 2024-11-22

I have read and agree with the venue's withdrawal policy on behalf of myself and my co-authors.